# Protein-Based Fat Replacers: A Focus on Fabrication Methods and Fat-Mimic Mechanisms

**DOI:** 10.3390/foods12050957

**Published:** 2023-02-23

**Authors:** Niloufar Nourmohammadi, Luke Austin, Da Chen

**Affiliations:** 1Department of Animals, Veterinary and Food Sciences, University of Idaho, Moscow, ID 83844, USA; 2Department of Biological Sciences, University of Idaho, Moscow, ID 83844, USA

**Keywords:** protein, fat replacer, fat mimic, microparticulation, microgels, thermomechanical treatment

## Abstract

The increasing occurrence of obesity and other non-communicable diseases has shifted the human diet towards reduced calorie intake. This drives the market to develop low-fat/non-fat food products with limited deterioration of textural properties. Thus, developing high-quality fat replacers which can replicate the role of fat in the food matrix is essential. Among all the established types of fat replacers, protein-based ones have shown a higher compatibility with a wide range of foods with limited contribution to the total calories, including protein isolate/concentrate, microparticles, and microgels. The approach to fabricating fat replacers varies with their types, such as thermal–mechanical treatment, anti-solvent precipitation, enzymatic hydrolysis, complexation, and emulsification. Their detailed process is summarized in the present review with a focus on the latest findings. The fat-mimic mechanisms of fat replacers have received little attention compared to the fabricating methods; attempts are also made to explain the underlying principles of fat replacers from the physicochemical prospect. Finally, a future direction on the development of desirable fat replacers in a more sustainable way was also pointed out.

## 1. Introduction

With the prevalence of obesity and obesity-associated chronic diseases, customers become increasingly aware of the calorie intake of their diet. According to the National Health and Nutrition Examination Survey (2017–2020), ~42% of U.S. adults aged ≥20 have obesity [1]. The Dietary Guideline for Americans, 2020–2025, encourages the public to consume low- or non-fat foods for healthier diets [2]. Reducing calorie intake by decreasing the amount of fat in food, especially saturated ones, has been considered one of the strategies to reduce the occurrence of obesity [3]. Nevertheless, the commitment to consuming low-fat foods or maintaining a low-fat diet remains challenging because of their deteriorated texture and sensorial properties compared to those of full-fat ones. Hence, the development of fat replacers that imitate not only the functional role of fat but also its sensory features is essential to improve the quality attributes of low-fat foods. Depending on the properties and manufacturing approaches, fat replacers fall into two categories: (1) fat substitute: typically, biomolecules or their degraded products with little or no calories, functioning similarly to fat. They fall into three main groups based on their sources, which are carbohydrate-based, which can hold water and impart a creamy texture close to fat (such as starches and gums), protein-based (such as egg white, milk, and whey), and fat-based fat replacers, which are too large to be digested with little contribution to calories (such as Caprenin as cocoa butter fat substitute and Olestra) [4]. As a well-known fat substitute example, Olestra is a product composed of sucrose, and hexa-, hepta-, and octa-esters of saturated and unsaturated fatty acids [5], but its application has been limited due to the risks of causing gastrointestinal side effects, such as abdominal cramping [6]; (2) fat mimetic (FM): ingredients that partially mimic the organoleptic properties of animal fat, includes mainly food hydrocolloids (gums, cellulose microfibrils, pectins), proteins, protein aggregates, protein–polysaccharides composites, and emulsion gels.

Protein-based fat replacers have received increasing attention. They boost the protein nutrition of food products with a low-calorie contribution. According to the dietary reference, adequate intakes of at least 0.8 g/kg body weight of high-quality protein for sedentary adults, the optimum amount of 1.2–1.8 g/kg body weight for adults with moderate activity, and 1.8–2.2 g g/kg body weight for adults with hypertrophy and strength training per day would be an ideal goal for health enhancement. When it comes to the health-related impact of a high-protein diet, protein intakes above the recommended amount within a certain range could limit the appearance of sarcopenia, and reduce the loss of muscle mass [7]. Due to their highly reactive features towards pH, temperature, ions, and enzymes, protein-based fat replacers can be tuned with distinct physiochemical properties for expanded food applications, such as in yogurt, cream cheese, salad dressings, and frozen desserts. Common sources of proteins to develop fat replacers include egg white protein, whey protein, gelatin, soy, pea, and zein [8]. Animal proteins are considered as higher quality because of the well-balanced amino acid profiles and high digestibility and bioavailability. For plant proteins, they commonly lack cysteine and methionine; however, this nutritional deficiency could be overcome by mixing different types of plant proteins [9]. In addition, due to the presence of anti-nutritional factors, such as trypsin inhibitors, phenolic compounds, phytates, cyanogenic compounds, lectins, and saponins, the digestibility of plant proteins can be reduced unless properly processed [10,11,12]. Furthermore, the incorporation of protein-based fat replacers levels up the protein content in foods, which enhances protein nutrition. The protein-based fat replacements have advantages over carbohydrate-based ones in terms of flavor interactions [13] and the amount of fat that could be replaced [14]. However, protein-based fat replacers may not be suitable to be incorporated into overprocessed food products [15,16] because of protein denaturation and interaction with other components (e.g., Maillard reaction), resulting in a loss of functionality and fat-like mouthful feelings [17].

The types of fat replacers, and their characterization and food applications have been reviewed systematically [12,18,19]. However, the approaches to develop protein-based fat replacers and the mechanism of their fat-mimic effects, which varies significantly with the source of proteins, their molecular weight, solubility, and surface chemistry, remain rarely summarized. This information is essential to design fat replacers with desirable functionality using the most appropriate approach. We attempt to cover the latest findings on those within the last five years. Future perspectives on the production of protein-based fat replacers with improved sustainability are also provided.

## 2. Types of Protein-Based Fat Replacers

### 2.1. Protein Concentrates and Isolates

Protein concentrates or isolates can be used directly as fat replacers (Figure 1). They may be slightly denatured during the manufacturing process [20]. The former contains ~30–80% protein, whereas the latter reaches 90–92% [21,22]. Extensive research has been conducted on developing low-fat dairy products including yogurt [23], ice cream [24], and cheese [25,26] using protein concentrates/isolates, especially whey protein due to its high compatibility with other dairy ingredients and a matching flavor profile [27,28]. A higher level (up to 6.8%) addition of whey protein could improve syneresis, yield stress, storage modulus (G’), viscosity, and creaminess of the low-fat yogurt with reduced serum separation compared to the control [29,30]. In low-fat cheese, the partial (3–8%) replacement of fat with whey proteins has been found to improve its hardness because of the formation of more compact structures [31].

### 2.2. Microparticulated Proteins

Protein microparticulation is a process of aggregation. The size of the particles commonly ranges within 0.1–10 μm, but larger ones >10 μm have also been reported [32]. Protein particles with a size larger than 5 µm can be detected by oral mucosa [33]; thus, a smaller size is preferable to provide the smoothening mouthful feelings. The most widely used microparticulated proteins is from whey proteins, which was patented in 1988 and later commercialized with the brand name Simplesse^®^ [34]. Some other proteins, such as soy protein, bovine serum albumin [35], egg white protein [36], gelatin [37], zein [38], wheat protein [39], pea protein, and potato protein [40], have also been used to produce fat replacers, as summarized in Table 1. Due to their spherical shape and size, they were claimed to mimic fat droplets and create a smooth and creamy mouthfeel through a “ball-bearing” mechanism, which will be discussed later.

### 2.3. Protein–Polysaccharides Hydrogel

Either proteins or polysaccharides have the capacity to form hydrogels. Protein-based hydrogels are mainly particle type, whereas those from polysaccharides are “thread or linear” type. When mixing the two, complexation could occur [57], which tailors protein functionality in foods by altering its surface chemistry and aggregating behavior [52,58,59] (Figure 1). Many studies have confirmed that the addition of polysaccharides prevented the coalescence and interaction between microparticulated proteins, either by shielding charged groups or by decreasing the collision rate between molecules through an increase in the viscosity of the system. The presence of polysaccharides can also bind a large amount of water to provide a creaminess sensation through the oral process [60]. Thus, a protein-based fat mimetic in combination with polysaccharides is usually formulated to replace fat to develop low-fat products [27,37,57]. The polysaccharides used in complexation with protein are gum arabic [61], pectin [52,62], alginate [63], and xanthan [64]. Nowadays, there is a growing trend to develop plant protein–polysaccharide hydrogel from peas [52], lentils [65], and soybeans [66] to increase their sustainability.

### 2.4. Microgel Particles

Protein-based microgel particles are novel types of fat replacers, with a size at nano- to micro-meter level [32]. They can be categorized into different types: fragments of protein hydrogels, protein assembles, emulsified gel droplets [67], and protein–polysaccharides coacervates [68] (Figure 1). The particles have the dual properties of particles and polymers, which endows ideal lubricating effects. Proteins can be used as the sole component of the microgel particles, and their elasticity depends on the types of proteins used and the interactions among protein molecules. In general, plant protein-based microgel particles have smaller elastic moduli compared to those of animal proteins due to the formation of less covalent bonds [69]. By changing environmental conditions, such as pH and ionic strength, the surface charge of protein molecules alters, which further affects their electrostatic interactions, resulting in distinct elasticity. For example, a 15-fold increment of elasticity of whey protein microgel particles was found at pH 3 and pH 5.5 compared to those at neutral pH [70]. Besides protein itself, other components could also be incorporated into microgel particles, such as polysaccharides and/or plant oils. Either of them contributes to the increased deformability of the particles. When plant oil was used, the system turns into an emulsion type, with Pickering emulsion being the typical example [53]. The emulsion-type protein-based microgels commonly have a more regular shape and better lubricating capacity than those of protein-only particles due to the formation of a lubrication film from the protein nanoparticles and base oils [71]. However, they are vulnerable to environmental change and long-term stability remains a challenge. Synthetic polymers or organic solvents might be used to decrease the surface tension or increase the stability of emulsion-type microgel particles, but they are not suitable for food applications.

## 3. Production of Protein-Based Fat Replacers

The way to fabricate fat replacers associates closely with their physicochemical properties and functionalities, such as particle size and shape, viscosity, gelling, emulsifying, foaming, and water-holding capacity. This further affects the color, texture, and other sensory characteristics of the final products. For proteins with high water solubility, such as whey proteins, thermal treatment to trigger protein aggregation under the controlled shear is commonly used. For those with limited water solubility, microparticulation could be achieved by breaking down large aggregates into smaller ones or improving their solubility/dispersibility first followed by precipitation. In the following sections, the common approaches used to produce fat replacers are discussed with a focus on protein microparticles and microgels.

### 3.1. Protein Concentrate or Isolate

The production of a protein concentrate or isolate has been well summarized [72,73,74,75] and will not be detailed here. Animal proteins, especially dairy proteins, are concentrated by ultrafiltration [26] or diafiltration followed by optional evaporation of the retentate prior to spray drying (Figure 2). The temperature of spray drying associates closely with the denaturation of the proteins, and their aggregation and particle morphology. The mean diameter of whey protein powders has been reported to increase from ~15 µm to ~20 µm when the outlet temperature was increased from 60 °C to 100 °C [76]. The morphology of whey protein concentrate particles was changed from spherical at a lower inlet air temperature to a deflated shape at high temperatures [77]. How the morphology and size of particles in a milk protein isolate or concentrate affect their fat-mimic capacity remains poorly studied. It has to be noted that the size and morphology of the particles in powders may change after they are incorporated into the food matrix due to the presence of water, salt, or other food components.

For plant proteins, wet and dry fractionation has been widely used (Figure 2). Wet methods include mainly alkaline-isoelectric point precipitation, or water, salt, or acid extraction. By using acid or alkaline, the protein recovery efficiency is high due to the increased solubility of proteins at the pH values far away from its isoelectric point [41,42]. Alkaline-isoelectric precipitation is more commonly used than other methods in food industries to extract proteins followed by spray drying to produce a protein isolate [78], but it uses a large amount of water and generates excessive salt in wastewater, which potentially threatens the freshwater ecosystem. In addition, high pH used during extraction may cause denaturation, racemization, and lysinoalanine formation of plant proteins, resulting in deteriorated quality [78,79]. Dry fractionation is more environmentally friendly and specific to produce a protein concentrate (40–50% protein content). It takes advantage of the size and charge of proteins that differed from those of starch and dietary fiber and achieves the separation by sieving or electrostatic interactions forces [72]. Since no high pH and extensive heat are involved during isolation, the proteins tend to have better functionality than those from the alkaline method.

### 3.2. Microparticulated Proteins

#### 3.2.1. Thermal–Mechanical Treatments

Thermomechanical treatment is the most common approach to develop microparticulated proteins as fat replacers. In general, a heating source is needed to unfold proteins by heating above their denaturation temperature (Figure 3). Unfolded proteins are then aggregated via covalent (S-S bonds) and non-covalent interactions (mainly hydrophobic interactions). Animal proteins are commonly heated at 75–95 °C for 20–40 min for microparticulation to occur [80,81,82]. More extensive heating (80–95 °C, ~30 min) is required for plant proteins due to their higher thermal stability [43,54]. The duration of heating shortens with the increase of temperature. For instance, pea protein particles were formed within a minute at 135 °C [45]. The morphology and size of the protein particle changes with the heating conditions. At lower temperatures, particles commonly have a smaller size with higher compactness [45,82,83,84]. Besides temperature, the shear force also applies to the system during heating unless at a low protein concentration (≤5%), otherwise, gelation could occur. The size of the particles negatively correlates with the shear force. High shear force results in more rigorous breaking of the aggregates and produces particles with smaller sizes [43], as demonstrated in whey proteins [82].

Multiple strategies have been adopted to provide heat and shear simultaneously (Figure 3). The simplest setup is to stir the sample on a temperature-controlled water bath. The strength of the shear can be fulfilled through adjusting the stirring speed, but the heating and cooling rate may be beyond control unless a sophisticated heating/cooling system is used. The other shortcoming of the method is it fails to provide ideal mixing of the concentrated protein suspension because of the high viscosity. These can be overcome by using a concentric cylinder accessory (bob-cup) equipped on a rheometer. The shear rate of the bob can be adjusted to the required value while the temperature is controlled by the cup. Such a method has been used to design microparticulated structures from whey, potato, and pea proteins [43] (Table 1). Within a certain range, the increase of the shear rate reduces the size of the formed particles as it disrupts the accumulated aggregation of proteins. The concentric cylinder is also able to monitor the microparticulation progress and explore the effects of shear force and temperature on the viscoelasticity, but the large-scale production of microparticles using a concentric cylinder remains a challenge as the volume of the cup is limited (e.g., 25 mL or less). The extrusion process, another thermomechanical treatment, is widely applied in food industries with the features of large-scale and continuous production. Proteins are unfolded and aggregated under the control of heating and screw rotation. Compared to the water bath and concentric cylinder, the temperature inside the extruder can achieve above 100 °C due to the high-sealed environment [45]. This facilitates protein microparticulation within a short duration. In addition, the protein content in the fed materials can be adjusted to higher values (≥20%), which significantly increases the yield of protein microparticles [85]. Whey protein particles produced through extrusion have been found to enhance the creaminess of low-fat, plain, stirred yogurt while maintaining its consumer acceptance with a less than 0.5% addition [44,85]. Similarly, the incorporation of pea protein microparticles derived from the same method into fat-reduced milk desserts resulted in approximately the same creamy perception as the full-fat milk dessert [45]. The conventional extrusion method consumes a large amount of energy to unfold proteins for subsequent aggregation. A hybrid technology, which takes advantage of the expanding properties of supercritical carbon dioxide (SC-CO_2_) to induce aggregation at lower temperatures (<100 °C), is more energy efficient [86]. In supercritical fluid extrusion, proteins tend to expose the hydrophobic regions and aggregates driven by the surrounding non-polar environment, resulting in distinct properties of microparticles, such as the protein–protein interactions, surface hydrophobicity, and rheological properties [87].

High-pressure homogenization combined with heat treatment is another technique capable of applying a strong shear force and a sudden pressure modulation to tune protein aggregation (Table 1). Liu et al. used such a method (10,000 rpm for 60 s, 75 °C for 13 min) to produce microparticulated egg white proteins as a fat replacer in salad dressings, which are comparable to the commercial salad dressings’ features [46]. Ultrasound-assisted heating could also induce protein microparticulation. The ultrasonication could be conducted simultaneously with the heating or after. A recent study found large whey protein aggregates (60–600 µm) were formed by heating, whose size was reduced to 0.01–2 µm upon sonication [47]. The treatment also resulted in higher surface hydrophobicity of the particles, which is possibly due to the exposure of more interior regions of protein aggregates.

#### 3.2.2. Enzymatic Hydrolysis

Protein aggregation and microparticulation could also be triggered by enzymatic treatment with or without additional heating. Transglutaminase catalyzes acyl transfer between ε-amino groups of lysine residues and the γ-carboxyamide groups to form covalent cross-links, but the reaction is slow and requires hours to complete, which hinders large-scale production [88]. The breaking down of proteins to expose hydrophobic regions or reactive amino acid residues also induces aggregation [48], which can be fulfilled through limited enzymatic hydrolysis. Compared to heat-induced aggregation, proteolysis is a greener approach, which consumes less energy. Depending on the types of proteases and the proteins, the conditions to promote aggregation vary. For instance, small aggregates were formed in *Bacillus licheniforms* (BLP) hydrolyzed whey proteins when 70% of the proteins were intact [89]. Using the same enzyme, whey proteins were found to form soft and turbid aggregate gels at 50 °C for 1 h [90]. For pea proteins, when the degree of hydrolysis was controlled to around 6–7% by a short time (2–3 min) hydrolysis with alcalase at 50 °C, the obtained hydrolysates were aggregated rapidly in response to heat. Analysis of the aggregates found that non-covalent interactions were the dominant forces that drive aggregation [48]. This suggests the aggregates might be deformed easily to provide lubricating effects. Zang et al. studied the influence of limited enzymatic hydrolysis of rice bran proteins by trypsin, at the ratio of 1:100 (Enzyme:Substrate)(*v/w*), on the emulsifying properties. They found a significant enhancement of the emulsifiying properties of hydrolysates with a 3% degree of hydrolysis. This was due to the release and exposure of soluble peptides from insoluble aggregates resulting in the exposure of more ionizable amino groups [91].

#### 3.2.3. Anti-Solvent Precipitation

Anti-solvent precipitation has been used to produce fine particles at the micro- and nano-scale level. The size and morphology of the particles could be well controlled by adjusting the protein concentration and polarity of the solvent [92]. The method is mainly applied to fabricate prolamin-based composites, such as zein and wheat gluten. Zein is the dominating protein found in maize, which can be extracted from dry-milled corn (DMC) or distillers-dried grains by using aqueous ethanol, acetic acid, or alkaline [93,94]. The hydrophobic nature of zein and its incomplete amino acid profile limit the food applications. Nevertheless, the inherent hydrophobicity and the heat-softening capacity of zein make it an ideal candidate for fat analog [95]. To achieve this, zein or its aggregates are suggested to micronize to a level similar to those of oil droplets by anti-solvent precipitation. Firstly, hydrophobic proteins are solubilized in an organic solvent, such as aqueous ethanol (70 to 90% *v/v*) or acetic acid/ethanol solution (e.g., at 55/45 ratio, *v/v*, pH 3.0) [96] under agitation [97,98]. Then, the concentration of the organic solvents is lowered down by adding water, resulting in increased polarity of the environment [99]. This triggers the aggregation of zein or wheat gluten mediated by hydrophobic interactions and precipitates out from the system when gravity overcomes electrostatic repulsions. Cui et al. have used zein nanoparticles prepared from anti-solvent precipitation as a stabilizer in low-fat emulsions. They solubilized zein in an 85% (*v/v*) aqueous ethanol at room temperature followed by the addition of phytic acid (PA) solutions to improve its stability. By stirring the mixture continuously for 30 min at 600 rpm, zein protein nanoparticles with a mean size of ~160 nm were formed [100]. The concentration of the organic solvent and the drying temperature directly links to the size and stiffness of the formed nanoparticles. Bisharat et al. (2018) found that the size of zein particles was increased from an average of 200–500 nm to 1 µm as the ethanol concentration was decreased from 90% to 70%. When the drying temperature was decreased from 55 °C to 40 °C and then to room temperature, the Young’s modulus of the particles was dropped continuously, corresponding to a higher flexibility and less resistance to stretching [101,102]. Zein could also form particles containing polysaccharides using anti-solvent precipitation. The particles have been shown to stabilize oil droplets as a potential fat replacer in sausages [49,50].

#### 3.2.4. Protein–Polysaccharides Hydrogel

When proteins and polysaccharides co-exist, depending on the environmental conditions and the concentration of each component, they behave distinctly. Thermodynamic incompatibility occurs mainly at high protein and polysaccharide concentrations due to their differed chemical nature and affinity towards solvent. Nevertheless, phase separation may not occur because of the high viscosity of the system. When gelation occurs fast, for instance, heating under a high temperature, the extent of phase separation or inhomogeneity of the gel would become less [60]. This results in higher consistency of the gel texture. Even though heating is not the sole condition to form a protein–polysaccharides hydrogel, it is the most common approach. Depending on the types of proteins and polysaccharides, they can both gel under heat, but not always. Most of the proteins are heat-sensitive and tend to unfold and aggregate to form a gel under heat, whereas polysaccharides could remained unchanged [103]. The proportion of the polysaccharides affects the rheological and fat-mimic capacity of the mixed gel and can be tuned accordingly. For instance, when mixing soy protein isolate and cellulose nanofibrils (CF), with the increasing of CF, the creaminess was increased [104]. If strong electrostatic interactions occur during the initial heating of proteins and polysaccharides, they can be gelled without extra heating. Since the complexation occurs rapidly, the protein or polysaccharides solution needs to be prepared separately before mixing. A high-pressure homogenization might be required to facilitate a homogenous distribution of polysaccharides and proteins [51]. Using this approach, Fan et al. (2020) developed oat β-glucan/marine collagen peptides mixed gels to replace the fat in sausage products [51]. Adjusting the environmental pH or ionic strength enables us to modify the surface charge of proteins, which further tunes the interactions between proteins and polysaccharides [103], and the textural properties of the gel. As non-covalent interaction are weak forces, the formed gels commonly possess high deformability against force and heat as desirable fat mimics.

### 3.3. Microgel Particles

For protein-only microgel particles, a hydrogel needs to be prepared followed by size reduction via homogenization or microfluidization to form micro- or nano-particles. Gelation could be conducted by using either the hot or cold method. Heat gelation occurs normally at 80–95 °C at a relatively high protein concentration (≥10%) for 20–30 min to induce protein aggregation and cross-linking (Figure 4). Cold gelation also requires heat to denature the protein first, followed by aggregating using calcium [105], salt [56], polysaccharides [55], or acid [106] at room temperature (Table 1). By changing the concentration of the reactants, the microgel particles display distinct physicochemical properties. For instance, when the concentration of CaCl_2_ was increased from 0.02 to 0.1 M, the size of whey protein microgel particles was decreased with the increment of their viscoelasticity [105]. When using salt, the concentration should be high enough to screen the surface charge of proteins to promote coacervation [56], but not cause the “salt in” effects. Similar to salt, the charged polysaccharides could also mediate the aggregation of proteins by reducing their surface charge. The hydrogen bonding between proteins and polysaccharides may also contribute and possibly play more significant roles than the electrostatic interaction on large-scale aggregation [107].

Besides protein-only or protein-polysaccharides microgel particles, there is increasing interest to fabricate emulsion-based ones (Figure 4). Proteins could be gelled [53] or denatured via heating (e.g., 90 °C for 20 min) [54] followed by emulsification with plant oil using homogenization. Heating enables the exposure of the buried hydrophobic regions of proteins for increased non-polarity favoring the stabilization of oil droplets. When the concentration of denatured proteins is too low to form gels, additional gelling steps are required, such as acidification [54] and complexation with polysaccharides [108]. In some other cases, proteins are mixed with oil in their native states to form emulsions with or without the assistance of an emulsifier, then heated or acidified to trigger the gelation of proteins [109,110]. Thermal gelation has been shown to deliver aggregates or heterogeneous microgels, whereas gelation by acidification resulted in spherical and more homogenous particles with a better fat-mimic capacity [110]. To facilitate industrial applications, the microgel particles can be harvested by centrifugation or membrane filtration followed by spray drying to form powders.

## 4. The Fat-Mimicking Mechanisms

An ideal fat replacer should possess lubrication, flow properties, and heat melting capacity [111] analog to fat. Unfortunately, protein-based fat replacers can hardly mimic all of them. Providing lubrication for the reduced friction of food products is the main function of a fat replacer. One of the theories to explain the changes in textures is the “ball-bearing” effect. It refers to the protein microparticles or microgels employing a rolling mechanism similar to “ball bearings” (Figure 5). When force is applied, the balls rotate and/or slide to decrease frictions of the surrounding matrix [112]. The surrounding matrix could be the neighboring protein networks or the boundary regime between proteins and non-protein components [113,114,115]. The shape and size of the protein microparticles allow them to entrain in the narrow space between the tongue and palate in the mouth and provide creaminess and the perception of smoothness. The spherical shape and size of microparticulated fat replacers ranging from 0.2 to 9 μm are the key factors in providing the ball-bearing effect. This mechanism has been claimed to be dominant for microparticulated whey proteins in liquid and semi-liquid model foods [113]. Besides size and shape, the interior compactness of the protein particles is also highly desirable for the ball-bearing effect to occur. Sarkar et al. (2017) found negligible changes in the microstructure and size of whey protein microgel particles after the tribology test, implying their resistance towards friction force [115].

For the particles with irregular shapes and larger sizes, such as aggregates from plant protein or protein hydrolysates, their deformation or even disintegration would contribute to a soft and/or smooth texture (Figure 5). Mechanical deformation of food through chewing and a biting motion in the oral cavity helps facilitate organoleptic perception, such as flavor release or the textural attributes of food. These movements, alongside the forces that the tongue makes by bouncing and excreting against the palate, could deform or even break the protein aggregates through disrupting non-covalent interactions [116]. Klost et al. (2020) have changed the proportion of soluble and insoluble pea protein aggregates in the fermented gels to mimic the texture of yogurt. They found the gels with higher insoluble aggregates showed higher flexibility and were easier to deform, showing smaller elastic moduli. After reducing the size of the aggregates by applying a larger intercycle strain, the system was converted from predominantly elastic to plastic behavior [58]. In a recent study, Chen and Campanella (2022) observed a substantial reduction of viscosity of pea protein hydrolysate gels under shear. Such shear-thinning behavior was partially due to the breaking down of the formed aggregates in addition to the realignment of the aggregates toward the shear flow direction [48].

Another assumption for the fat-mimic mechanism is the emulsification effects where the fat replacers may form an emulsifier layer or oil layer on the respective surfaces for lubrication, depending on the stability of emulsion gel microparticles (Figure 5). For those with high stability [117], the oil droplets are entrapped by interacting with the inner surface of protein particles during mastication. The outer surface of protein particles contacts with other macromolecules (e.g., starch, proteins) in the food matrix. Since the oil droplet inside the emulsion gel microparticles has a low glass transition and high flexibility, the outer layer that made up of protein particles can act as a filler. When experiencing force such as chewing, the particles slide to each other and deform to reduce the friction and increase the creaminess. For emulsion-type particles with limited stability, during mastication, they can be broken down into small fragments at a small deformation with the release of entrapped oil. The released oil lubricates the foods and provides a smooth mouth perception [118]. For a certain type of fat replacers, a set of fat-mimic mechanisms may occur which requires a comprehensive assessment [32,119].

## 5. Conclusions and Future Trends

In summary, designing protein-based fat replacers is essential to improve the quality attributes of low-fat and fat-free food products; although, they are more costly compared to those of carbohydrate-based ones. Microparticulated proteins, as the dominant fat replacers, are produced mainly by thermal–mechanical treatment. Other fabrication methods, such as anti-solvent precipitation and enzymatic hydrolysis, have also been used with less heat requirement. Microgel particles as a novel type of fat replacement receives increasing attention. With the incorporation of plant oil, it could provide a better lubrication effect and creaminess. The mechanisms of protein-based fat replacers remain unambiguous. The ball-bearing effect, protein aggregates/particle deformation and disassembly, and their emulsification contribute significantly to the lubrication and flow properties of fat-replacer incorporated food products for desirable texture and mouthful feeling.

Although much progress has been made in the production, modification, and/or application of fat replacers, the development of novel protein-based fat replacers in a greener way is still in its infancy. The thermal–mechanical treatment requires a large amount of energy input to trigger protein unfolding and aggregation, which, in turn, levels up the cost of the final products and is not environmentally friendly. The modification of the protein structure using processing or bioprocessing to promote their aggregating capacity under heat requires further investigation. Most of the protein-based fat replacers do not have the heat melting property except the prolamin-based ones. The incorporation of zein particles to those of others with high compatibility and ideal fat-mimic effects is worthwhile to explore. There is a growing trend of using plant proteins as fat replacers and each of them has its own structural, physiochemical, and mechanical properties. Yet, the maximum amount of fat that can be replaced with plant-based protein is ambiguous. The evaluation of different protein sources, as well as combining plant-based with other animal-based proteins, such as whey, or other non-protein ingredients, such as carbohydrates [38], to test their fat-mimicking potential, needs further attention. In addition, tribology combined with rheological textural analysis is dominant to assess the lubrication of fat replacer-incorporated low-fat products [119]. The difference between instrumental analysis and human mouth sensing implies a combination of the two is more accurate to reflect the true fat-mimic capacity of protein-based fat replacers and needs focused attention.

## Figures and Tables

**Figure 1 foods-12-00957-f001:**
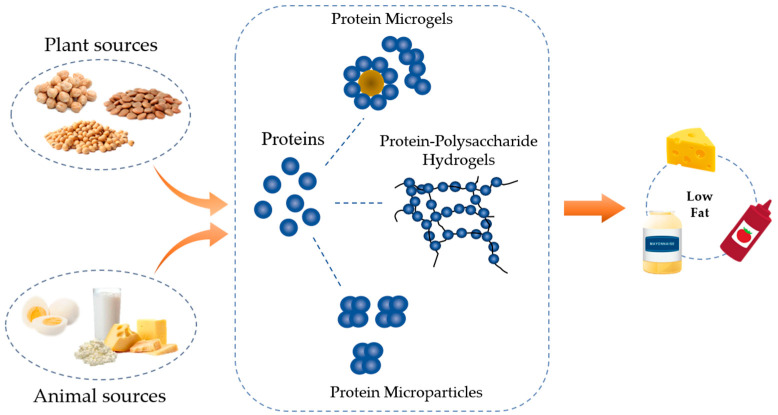
A schematic showing the types of protein-based fat replacers.

**Figure 2 foods-12-00957-f002:**
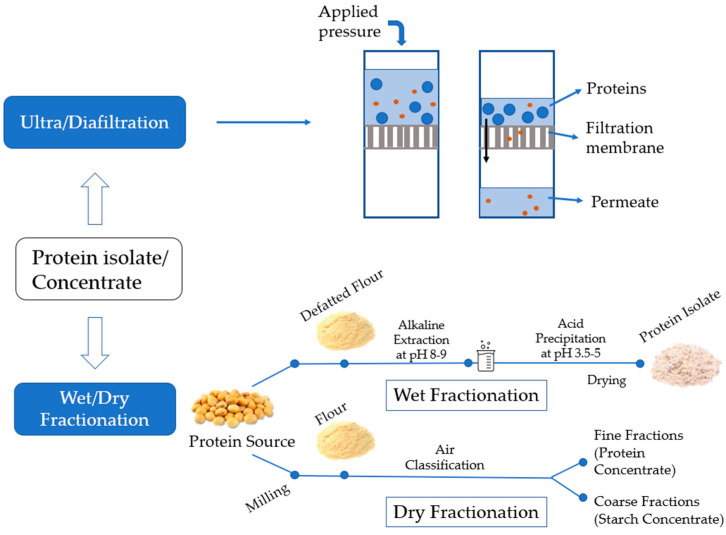
A schematic showing the production of protein isolates and concentrates.

**Figure 3 foods-12-00957-f003:**
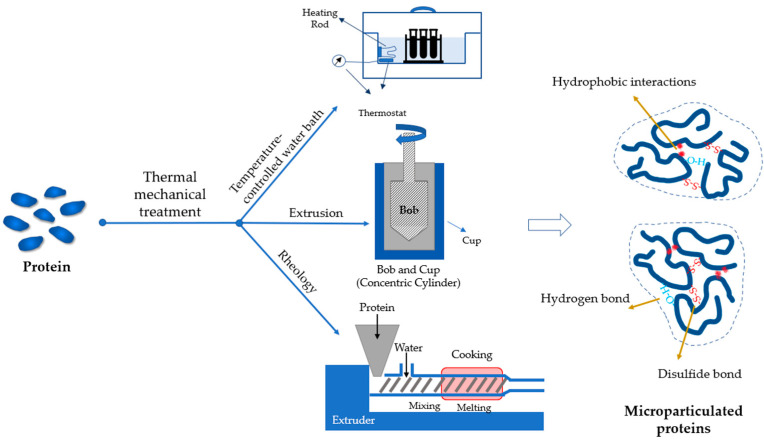
A schematic illustrating the production of microparticulated proteins using thermal–mechanical treatments.

**Figure 4 foods-12-00957-f004:**
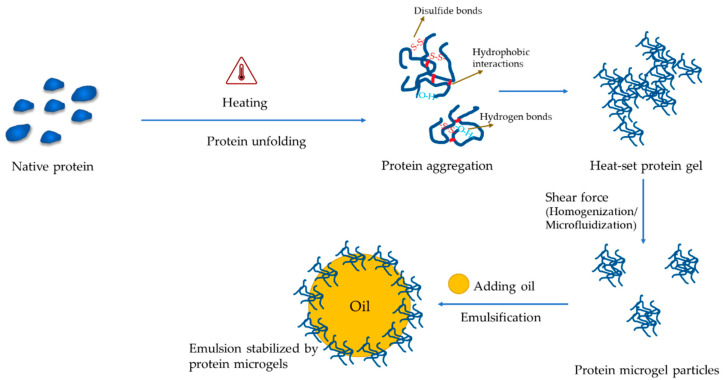
A schematic showing the formation of protein-based microgels.

**Figure 5 foods-12-00957-f005:**
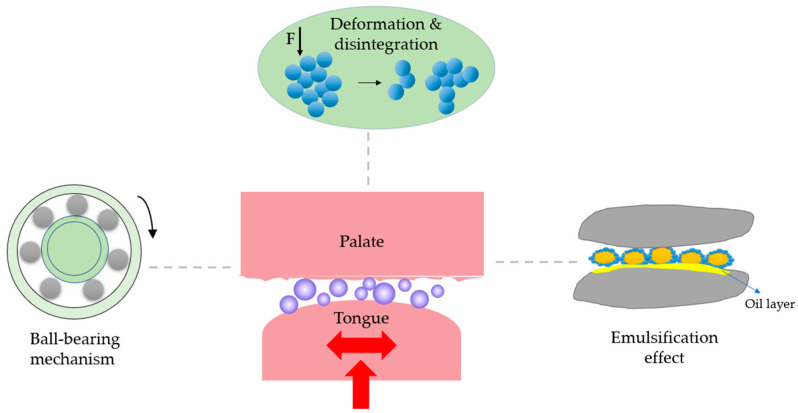
A schematic showing the mechanisms of fat-mimic effects.

**Table 1 foods-12-00957-t001:** A summary of different types of protein-based fat replacers, their fabrication methods, and applications.

Type	Protein Source	Fabrication Method	Particle Size (μm)	Application	References
Protein concentrate/isolate	Whey concentrates	Ultrafiltration at 40–45 °C, membrane cut-off 10 kDa	-	Reduced fat cheese	[26]
	Soy protein fractions	Protein solubilization at pH 8 and 9 for 1 h, separate oil-rich cream by centrifuge, protein precipitation in pH 4.5 and 5 by adding 1 M HCl hold for 1 h	10–250	Meat analog	[41,42]
Protein microparticles	Microparticulated soy protein/egg white protein	Heated 95 °C for 5–15 min with continuous stirring	2–20	Drinkable or semi-solidprotein-rich foods	[36]
	Microparticulated whey, potato, and pea proteins	Heating to 10 °C above denature temperature of proteins, held for 10 min, and cooling in a concentric cylinder with 100–150 s^−1^ shear rate	20–250	-	[43]
	Microparticulated whey proteins	Extrusion at 90 °C with a screw speed of 200–1000 rpm	2–7	Reduced-fat yogurt	[44]
	Potato protein	Extrusion at 80 °C and 800 rpm screw speed, pH 6.9	9–110	Fat-reduced dessert	[40]
	Pea protein	Extrusion cooking at 100 °C with 600 rpm screw speed	10–75	Milk dessert	[45]
	Egg white protein	Heated at 75 °C for 13 min, followed by high-shear homogenization at 10,000 rpm for 60 s	9.4	Salad dressing	[46]
	Microparticulated whey protein	Heated at 85 °C for 15 min and sonicated at 20 kHz for 1 min	0.01–2	Reduced-fat cheese emulsion	[47]
	Soy protein hydrolysate	Alcalase 2.4 L hydrolysate followed by heating at 90 °C for 20 min and homogenized at 8000 rpm for 6 min	7.1–9.3	Ice cream	[24]
	Pea protein hydrolysate	Hydrolysis by Alcalase 2.4 L for 3 min, followed by heating at 85 °C for 10 min	-	-	[48]
	Zein/ carboxymethyl dextrin	Zein was dissolved in 80% ethanol and then added to a dextrin solution, followed by the removal of ethanol	0.1–0.6	Sausage	[49,50]
Protein–polysaccharide complex	Oat β-glucan/marine collagen peptide	Oat β-glucan was dissolved in buffer solution at pH 3 (glycine–HCl buffer) at 12% concentration and mixed with marine collagen peptide solution at different ratios. The samples were stirred at 25 °C for 4 h, then subjected to high pressure at 400–500 MPa for 30 min	-	Sausage	[51]
	Pea protein/pectin	Pea protein powder was added to solutions containing different ratios of pectin to reach 15% (*w/v*) of protein concentrate. The dispersion was stirred at 1500 rpm for 1 h at room temperature followed by adjusting the pH to 6.5 using 2 N NaOH	-	-	[52]
Protein microgels	Canola protein microgels	Heated at 90 °C under stirring for 1 h, stored at 4 °C to form a gel followed by homogenization and complexation with polysaccharides	1–100	Pickering emulsion stabilization as a potential fat replacer	[53]
	Whey protein emulsion gel microparticles	Heated at 90 °C for 20 min, followed by the addition of oil, homogenization, and addition of glucono-delta-lactone	0.3–300	Low-fat yogurt	[54]
	Whey protein/alginate microgels	Whey protein was heated at 80 °C for 30 min followed by alginate addition	-	-	[55]
	Soy protein microgels	Salt-induced coacervation followed by heating at 80 °C for 30 min under stirring	1–3	-	[56]

Note: The “-“ means “it is not reported”.

## Data Availability

Not applicable.

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
