# Peer review of "Protein-Based Fat Replacers: A Focus on Fabrication Methods and Fat-Mimic Mechanisms"

_foods, 2023, doi:10.3390/foods12050957_

Round 1

Reviewer 1 Report

The authors reported “Protein-based Fat Replacers: a focus on fabrication methods and fat-mimic mechanisms". The types of fat replacers, their characterization, and their food applications have been reviewd systematically. The review will help better understand the high-quality fat replacers (FRs), which could replicate the role of fat in the food matrix are essential.

For the benefit of the reader, however, a number of points need clarifying and certain statements require further justification. Some detailed information is provided following. 

Specific comments: 

1. The second author's address shows duplicates. Deleted “University of Idaho”.

2. What are the common fat replacers?

3. Where does “Protein concentrates and isolates” usually obtained? Please outline the production process of protein concentrate or protein isolate

4. The approach to fabricate fat replacers varies with their types, such as thermal treatment, antisolvent precipitation, enzymatic hydrolysis, complexation and emulsification. Please compare the advantages and disadvantages of these methods.

5. The author attempt to cover the latest findings within the last five years. Please cite the recent references, especially for the part on “Microparticulated proteins”.

6. In page 2 line 63, "FRs" needs to be given a specific meaning for the first occurrence.

7. In page 5 line 226, What are the most commonly used antisolvent precipitates in production?

8. Please write properly in this paper. Such as 95 ◦C was changed to 95 ℃ in Table 1.

Reviewer 2 Report

The authors gave a summary on the protein-based fat replacer on the fabrication methods and fat-mimic mechanisms, which is an interesting topic that might provide some hints for related researchers. However, the manuscript is not well presented based on previous research. Here are my suggestions:

1.     Keywords should be revised. Rheology and tribology were not discussed extensively in the manuscript.

2.     Illustrations of the various fabrication methods for protein isolate/concentrate, protein microparticles, and protein microgels are essential in the respective sections. The main message of the review is to convey to the readers the way to fabricate protein based fat replacer. So the authors should summarize it to make it more clear.

3.     Similar to the illustration of the fabrication methods, Table 1 should be separated into sections as Table 1, Table 2, and Table 3 for protein isolate/concentrate, protein microparticles, and protein microgels sections, respectively. The tables should be supplemented according to references.

4.     There are also some format problems, the authors should check the manuscript carefully. 

Reviewer 3 Report

The review summarizes the main sources, types, production methods and the fat-mimicking mechanisms of proteins used as fat replacers. Several points need to be considered. 

1. What are the positive and negative consequences of using proteins as fat replacers on the human health? higher protein content could cause some health issues. 

2. In terms of economic evaluation, any edible food protein is an essential and expensive food element for humans' nutrition; so is it costly to use protein as a fat replacer compared to other fat replacers? 

3.  Please add a paragraph to compare proteins with other fat replacers. 

Round 2

Reviewer 2 Report

The revised version is acceptable.